Molecular Biology and Physiology

# Characterizing the Mechanism of Action of an Ancient Antimicrobial, Manuka Honey, against *Pseudomonas aeruginosa* Using Modern Transcriptomics

Daniel Bouzo,[a] Nural N. Cokcetin,[a] Liping Li,[b] Giulia Ballerin,[a] Amy L. Bottomley,[a] James Lazenby,[a] Cynthia B. Whitchurch,[a] Ian T. Paulsen,[b] Karl A. Hassan,[c] Elizabeth J. Harry[a]

[a]The ithree institute, University of Technology Sydney, Ultimo, NSW, Australia
[b]Department of Molecular Sciences, Macquarie University, North Ryde, NSW, Australia
[c]School of Environmental and Life Sciences, University of Newcastle, Callaghan, NSW, Australia

**ABSTRACT** Manuka honey has broad-spectrum antimicrobial activity, and unlike traditional antibiotics, resistance to its killing effects has not been reported. However, its mechanism of action remains unclear. Here, we investigated the mechanism of action of manuka honey and its key antibacterial components using a transcriptomic approach in a model organism, *Pseudomonas aeruginosa*. We show that no single component of honey can account for its total antimicrobial action, and that honey affects the expression of genes in the SOS response, oxidative damage, and quorum sensing. Manuka honey uniquely affects genes involved in the explosive cell lysis process and in maintaining the electron transport chain, causing protons to leak across membranes and collapsing the proton motive force, and it induces membrane depolarization and permeabilization in *P. aeruginosa*. These data indicate that the activity of manuka honey comes from multiple mechanisms of action that do not engender bacterial resistance.

**IMPORTANCE** The threat of antimicrobial resistance to human health has prompted interest in complex, natural products with antimicrobial activity. Honey has been an effective topical wound treatment throughout history, predominantly due to its broad-spectrum antimicrobial activity. Unlike traditional antibiotics, honey-resistant bacteria have not been reported; however, honey remains underutilized in the clinic in part due to a lack of understanding of its mechanism of action. Here, we demonstrate that honey affects multiple processes in bacteria, and this is not explained by its major antibacterial components. Honey also uniquely affects bacterial membranes, and this can be exploited for combination therapy with antibiotics that are otherwise ineffective on their own. We argue that honey should be included as part of the current array of wound treatments due to its effective antibacterial activity that does not promote resistance in bacteria.

**KEYWORDS** *Pseudomonas aeruginosa*, RNA-Seq, antimicrobial activity, honey, manuka honey, mechanism of action, natural antimicrobial products, transcriptomics

Address correspondence to Nural N. Cokcetin, nural.cokcetin@uts.edu.au.

Honey has been used for millennia as a topical antibacterial (1–6), and, unlike traditional antibiotics, bacterial resistance to honey has not been reported (7, 8). The increasing prevalence of antimicrobial resistance demands alternative infection control and has prompted renewed scientific interest in complex, natural products with potent antimicrobial activity, like honey. However, honey remains underutilized in the clinic, presumably due to a paucity of information on the mechanisms by which honey kills bacteria.

Honey is a complex mixture, with over 100 components, including sugars, proteins,

phenols, and plant- and bee-derived enzymes (9). The antibacterial activity of honey is derived from multiple factors: osmotic stress from the high sugar concentration (10, 11), low pH (between 3.2 and 4.5), and the presence of hydrogen peroxide ($H_2O_2$) produced from the bee-derived enzyme glucose oxidase. It was widely considered that the latter was the primary source of the antibacterial activity of honey, and it is known to vary significantly in honeys from different floral sources (10, 12–15); however, following the neutralization of $H_2O_2$ by catalase, certain honeys retained high levels of antibacterial activity, referred to as nonperoxide activity (NPA). NPA was first observed in New Zealand manuka (*Leptospermum scoparium*) honey (MH) (12, 13). It has now been established that active manuka-type (*Leptospermum* sp.) honeys from New Zealand and Australia have substantially higher levels of NPA than honeys from other floral sources (14, 15). This is due, in part, to the high concentrations of the naturally occurring chemical methylglyoxal (MGO) in some *Leptospermum*-derived honeys (16, 17).

While MGO is a key antibacterial component of manuka honey, it alone cannot account for its total antimicrobial activity (18–20), as manuka honey inhibits the growth of pathogenic bacteria (including *Pseudomonas aeruginosa*, *Escherichia coli*, and *Staphylococcus aureus*) in their planktonic and biofilm lifestyles at concentrations well below the MIC of MGO alone (18–22). Additionally, many bacteria are innately equipped to detoxify MGO (23–25), so additional components in honey must also modulate its activity. From this, we hypothesize that the antibacterial activity of manuka honey comes from a combination of its various constituents and that its mechanism of action cannot be elucidated based exclusively on investigations of the individual components. Rather, to generate a fundamental understanding of the mechanism of antibacterial activity, the effects of the key components of manuka honey against microorganisms must be studied in isolation from and in combination with each other. Despite the prominent role of MGO in the antibacterial activity of manuka honey, the degree to which it contributes to the effect manuka honey has on bacterial gene expression and physiology has not been thoroughly investigated (26–32). Currently, the antimicrobial activity of manuka honey is reported and marketed based on its NPA, which can be directly tested via bioassays or derived from the MGO concentrations of manuka honey, since MGO and NPA are well correlated (15). This is problematic, since NPA is only a measure of antistaphylococcal activity and not representative of activity against other bacterial species (33). Therefore, it is important to understand how MGO alone and in combination with sugars works against Gram-negative microorganisms like *P. aeruginosa* to better understand the mechanism of whole manuka honey. This is critical for its use in infection control, which requires the killing of multiple species of bacteria present in wounds.

Previous studies have identified a number of biological processes in bacteria that may be affected by the action of honey, including cell division (19, 27, 30, 31), motility (26), quorum sensing (QS) (34–38), protein synthesis (27, 30, 81), and responses to oxidative stress (7, 36). With the increased affordability, sensitivity, and accessibility of genetic analysis, we can now elucidate the entire changes that happen to a bacterial cell when exposed to different treatments. We have used a global transcriptomic approach, transcriptome sequencing (RNA-Seq), as well as classical cell biology techniques, to characterize the effects of manuka honey and its key components (MGO, sugar, and their combination) on *P. aeruginosa*. This opportunistic pathogen is commonly associated with burn wounds and surgical site infections (82) and is listed by the World Health Organization as a Priority 1 critical pathogen for which novel treatment therapies are urgently needed (83). We demonstrate that (i) exposure to manuka honey causes significant, widespread changes in the transcriptomic profile of *P. aeruginosa*; (ii) the mechanism of action and effect of honey on the transcriptomic response of *P. aeruginosa* is different from that of MGO, sugar, or a combination thereof; and (iii) only whole manuka honey, and not these key components, dissipates membrane potential in *P. aeruginosa* and is an important part of the mode of action of manuka honey.

**TABLE 1** MIC and MBC values of manuka honey and honey analogues against *P. aeruginosa* PA14[a]

| Sample[b] | MIC (% wt/vol ± SD) | MBC (% wt/vol ± SD) | MGO content (ppm) at MIC[c] |
|---|---|---|---|
| MH | 10 ± 0.25 | 12 ± 0.25 | 90 |
| AH | 25 ± 0.5 | >30 ± 0.00 | 0 |
| MGO | 55 ± 0.00 | >55 ± 0.00 | 495 |
| AH + MGO | 21.5 ± 0.5 | >30 ± 0.00 | 193.5 |

[a]Minimum inhibitory concentration (MIC) and minimum bactericidal concentration (MBC) are expressed as the mean percentages (wt/vol) ± SD from three separate trials, all performed in triplicate. Percent concentrations for manuka honey components are presented relative to their concentration within whole manuka honey.
[b]MH, manuka honey; AH, artificial honey; MGO, methylglyoxal.
[c]Quantity of MGO in parts per million (ppm) present at MICs of each sample.

## RESULTS AND DISCUSSION

**The antimicrobial activity of manuka honey against *P. aeruginosa* cannot be explained solely by methyglyoxal presence or levels.** We determined the contribution of MGO and sugar (either in isolation or in combination with one another) to the antibacterial activity of manuka honey against *P. aeruginosa* by minimum inhibitory and minimum bactericidal concentration (MIC and MBC, respectively) assays (Table 1).

*P. aeruginosa* growth was inhibited by 10%, wt/vol, manuka honey, in agreement with previous reports (39), and manuka honey was bactericidal at 12%, wt/vol (Table 1). The MIC values for artificial honey (AH), AH and MGO combined (AHMGO), and MGO alone were higher than that of manuka honey, and MGO had the highest MIC of all treatments; none of these treatments were bactericidal at the highest concentrations that could practicably be tested (Table 1). The MIC for MGO alone was 5.5-fold higher than that of manuka honey (equivalent to 495 ppm MGO compared to 90 ppm MGO, respectively), similar to previous reports for the MIC of MGO against *P. aeruginosa* PAO1 (22). These results show that although MGO does contribute to the activity of manuka honey against *P. aeruginosa*, it is not the main factor responsible for inhibition or cell death. This is very different from *S. aureus*, where the MGO content of manuka honey correlates strongly with antistaphylococcal activity (15).

**Manuka honey affects a range of biological processes and pathways.** The molecular responses of *P. aeruginosa* to manuka honey and its key components were investigated using RNA-Seq. We applied treatments at subinhibitory concentrations and short exposure times (0.5× MIC for 30 min), since this approach induces more specific responses and reduces indirect effects, thereby giving the most informative transcriptomic data (40, 41). We confirmed these conditions induced significant and meaningful changes in gene expression by pilot reverse transcription-quantitative PCR (RT-qPCR) experiments on targeted genes (see Fig. S1 in the supplemental material). It should be noted that using 0.5× MIC across all treatments meant that the final MGO concentration was different under each treatment condition in the RNA-Seq experiments (Table S2), and we were mindful of this in interpreting the data.

We first explored the transcriptomic changes in *P. aeruginosa* induced by manuka honey. Manuka honey markedly affects the transcriptomic profile of *P. aeruginosa* compared to that of the untreated control (Fig. 1A), with changes to the expression of 3,177 of 5,892 coding sequences (54%; false discovery rate threshold of 0.05). A similar number of genes was upregulated ($n = 1,646$, representing 28% of all coding genes) versus downregulated ($n = 1,531$, or 26%). Analysis of only the genes with a $\log_2$ fold change ($\log_2$FC) of $\geq \pm 2$ showed that 235 were differentially expressed, equivalent to 4% of all coding sequences. When this thresholding was applied, more genes were upregulated than downregulated (Fig. 2A). Principal component analysis (PCA) confirmed that the effect of manuka honey on *P. aeruginosa* differed markedly relative to the untreated control (Fig. 1B).

Genome-wide expression changes were visualized as volcano plots (Fig. 2) to identify specific genes with large fold changes and statistical significance. Genes that were significantly differentially expressed (adjusted *P* value [$P_{adj}$] < 0.05) and above a

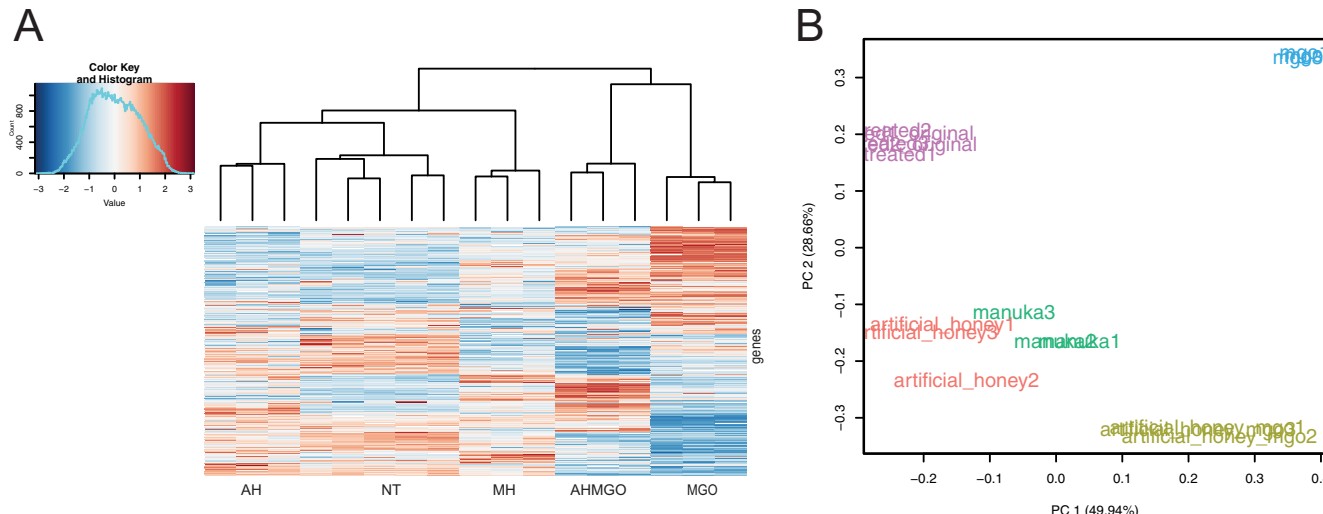

**FIG 1** Transcriptional response of *P. aeruginosa* PA14 treated at mid-exponential phase with manuka honey and honey analogues for 30 min at 0.5× MIC. (A) Clustered heatmap of relative expression of 5,892 coding genes in *P. aeruginosa* across all treatments and a no-treatment control (NT). Treatments include whole manuka honey (MH) and its constituents: artificial honey (AH), artificial honey doped with methylglyoxal (AHMGO), and methylglyoxal (MGO) alone. RNA-Seq was performed using three biological replicates for each treatment and five for the no-treatment control. The clustered heatmap shows the row z-score (amount by which counts for a gene deviates in a specific sample from that gene's average across all samples) and is clustered based on Euclidean measures and complete agglomeration. (B) Bi-plot of the principal-component analysis of normalized read counts for all treatments (manuka honey, green; MGO, blue; artificial honey, red; artificial honey plus MGO, tan) and the untreated control (purple), split into biological replicates.

$\log_2$FC of >2 are presented in red, and the five most up- and downregulated genes are labeled in each plot. Manuka honey treatment caused significant upregulation of 138 genes and downregulation of 97 genes (Fig. 2A). In the manuka-treated sample, the two genes in the PA14_56360-56370 operon were among the top five most upregulated ($\log_2$FC of 5.87 and 5.85, respectively). These genes encode hypothetical proteins that share homology with proteases from the DJ-1/PfpI family in *P. aeruginosa* PAO1, namely, the oxidative stress response gene *ahpF* ($\log_2$FC = 5.46), the glyoxalase enzyme *gloA3* ($\log_2$FC = 5.19), and the aldo-keto reductase (AKR) *gsp69* ($\log_2$FC = 5.16) (Fig. 2A). The genes that had the largest downregulation following manuka treatment were the *scoAB* operon, encoding coenzyme A transferase subunits A and B ($\log_2$FC = −4.94 and −4.15, respectively), *hmgA* ($\log_2$FC = −3.97), encoding a homogentisate 1-2-dioxygenase, and a pair of genes in the PA14_58410-58490 operon, encoding the products of the outer membrane porin *opdP* (PA14_58410) ($\log_2$FC = −3.79) and periplasmic ABC transporter *dppA4* (PA14_58420) ($\log_2$FC = −3.62) (Fig. 2A).

Differential expression of genes related to certain biological functions defined by PseudoCAP (42) was also analyzed (Fig. 3). PseudoCAP categorizes genes based on experimental evidence of their involvement in a particular cellular function or their assignment to KEGG pathways participating in that function. The percentage of genes in each PseudoCAP classification that are differentially expressed ($\log_2$FC ≥ ±2 and $P_{adj}$ ≤ 0.05) was used as an indication of the extent to which manuka honey affected particular biological functions in *P. aeruginosa*.

Overall, manuka honey had a major effect (upregulation) on the following categories: related to phage, transposon or plasmid; antibiotic resistance and susceptibility; and adaption and protection (Fig. 3). Other categories upregulated by manuka honey in a sizeable but smaller manner were those relating to the transport of small molecules; transcription, RNA processing, and degradation; secreted factors; putative enzymes; nucleotide biosynthesis metabolism; membrane proteins; energy metabolism; DNA replication, recombination, modification, and repair; central intermediary metabolism; and cell wall and biosynthesis of cofactors. In comparison, only a few processes were downregulated by manuka honey, and these were protein secretion, fatty acid and phospholipid metabolism, and amino acid biosynthesis and metabolism (Fig. 3).

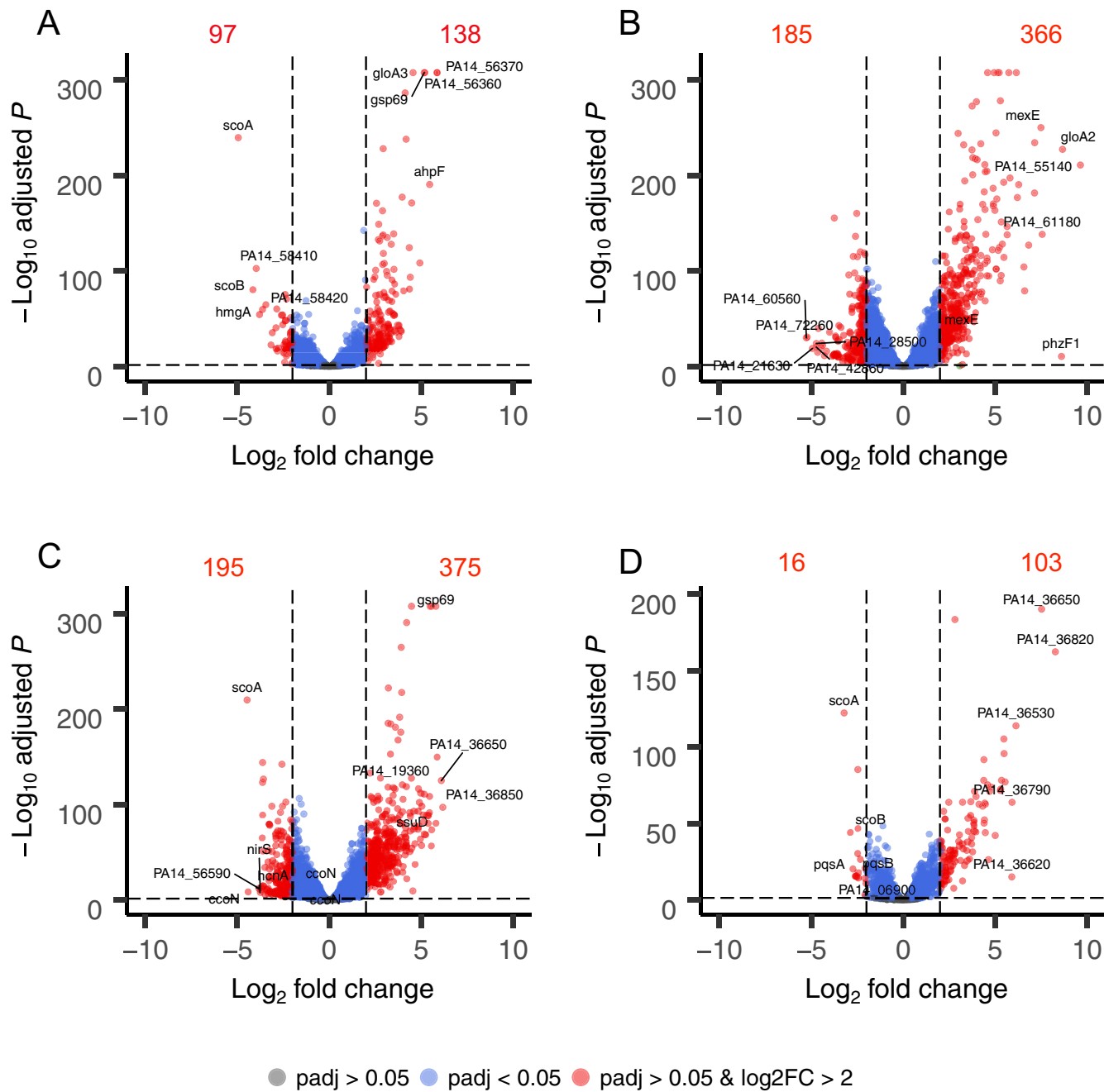

**FIG 2** Summary of genome-wide expression changes in *P. aeruginosa* PA14 after treatment with manuka honey (A); MGO (B); AHMGO (C); and AH (D). The top 10 most differentially expressed genes are labeled in each plot. Gray dots indicate genes with no significant difference compared to the untreated control ($P_{adj} > 0.05$), blue dots indicate genes with a significant difference compared to the untreated control ($P_{adj} < 0.05$), and red dots indicate genes with both a significant difference ($P_{adj} < 0.05$) and $\log_2$FC of >2 compared to the untreated control, with numerical annotations to indicate the number of differentially expressed genes.

Functional groups corresponding to biological processes were manually curated and visualized as heatmaps (Fig. 4). Functional groups were selected where several genes involved in a particular pathway or process were affected and where at least two of those genes were among the 25 most differentially up- or downregulated. Manuka honey treatment induced the differential expression of genes involved in (but not limited to) quorum sensing (Fig. 4A), the oxidative stress response (Fig. 4B), and the SOS response (Fig. 4C) and tailocin (sometimes referred to as pyocin) genes (Fig. 4D) (this is discussed below). To our knowledge, this is the first report of SOS induction by manuka

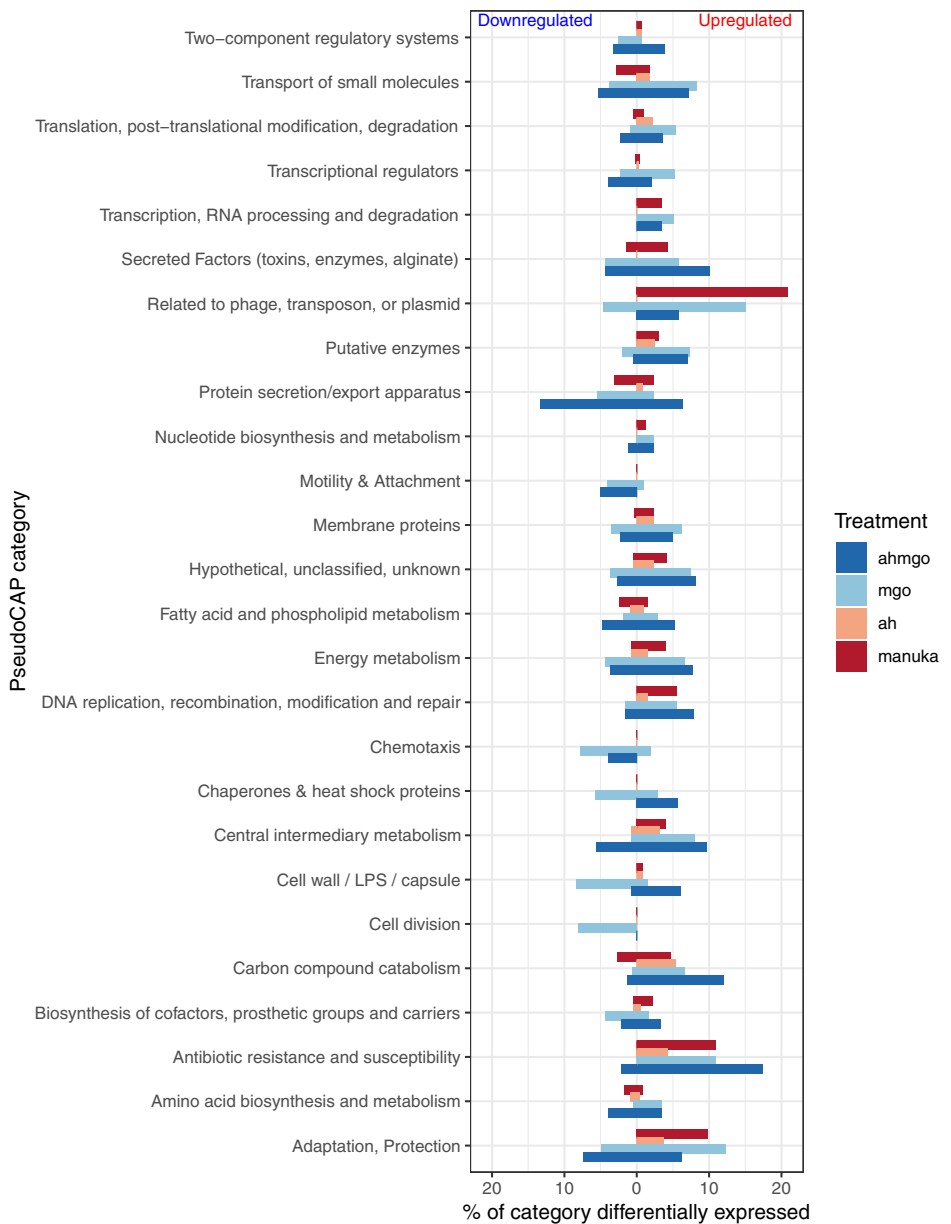

**FIG 3** Percentage of genes of each PseudoCAP category that were differentially expressed (log$_2$FC of ≥±2 and $P_{adj}$ ≤ 0.05) after treatment with manuka honey (dark red), MGO (light blue), AHMGO (dark blue), and AH (light red). LPS, lipopolysaccharide.

honey in any microorganism. Our data suggesting that manuka honey affects quorum sensing via the downregulation of the *pqsABCDE* operon supports previous studies in *P. aeruginosa* PAO1 (35). Complementary techniques, such as microarray analysis, genetic screens, and proteomic approaches (27, 43–45), have shown honey can affect the expression of genes involved in the oxidative stress responses in *S. aureus* and *E. coli*, and our findings indicate that this also occurs in *P. aeruginosa*.

To explore whether oxidative stress responses were due to the generation of reactive oxygen species (ROS), thought to be a common killing factor for many antimicrobials, we performed MIC assays under anaerobic conditions where ROS formation is impeded (46–48). There was no difference in the MIC of manuka honey under aerobic (MIC, 10%, wt/vol) versus anaerobic (MIC, 11%, wt/vol) conditions ($P > 0.05$), suggesting that ROS (and related oxidative stress) is not the only contributor to the

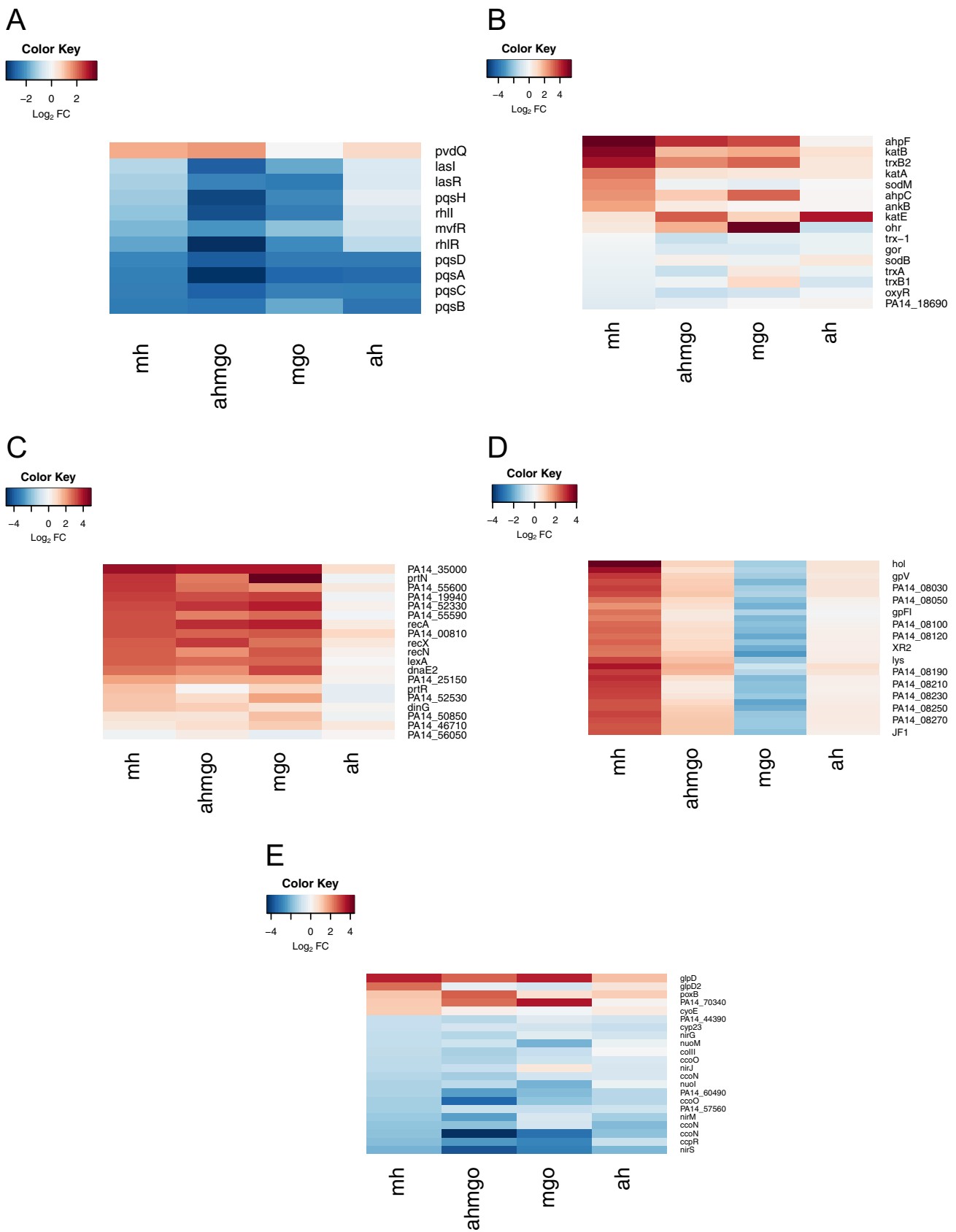

**FIG 4** Heatmaps show log$_2$FC data for *P. aeruginosa* PA14 treated at mid-exponential phase with manuka honey and honey analogues for 30 min at 0.5× MIC. Quorum-sensing genes (A), oxidative stress response genes (B), SOS response genes (C), tailocin genes (D), and respiration genes (E) for each treatment.

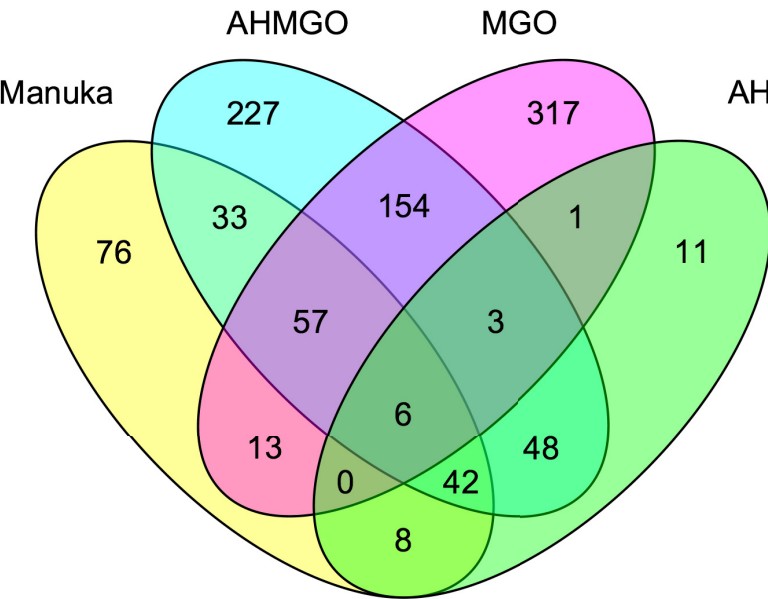

**FIG 5** Venn diagram of differentially expressed genes (log$_2$FC $\geq$ 2, $P_{adj} \leq$ 0.05) in *P. aeruginosa* PA14 after treatment with (yellow) manuka honey (yellow), AHMGO (blue), MGO (pink), and AH (green) at 0.5$\times$ MIC.

antimicrobial mechanism of action. We have also previously demonstrated that exponentially growing *P. aeruginosa* PAO1 cells had condensed chromosomes after treatment with 4%, wt/vol, manuka honey, suggesting that it inhibits DNA replication in these cells (19). DNA degradation by oxidative damage would result in dispersed chromosomes rather than condensed ones, indicating that oxidative stress is not the mechanism of death in manuka honey-treated *P. aeruginosa*.

**Can the transcriptomic effects of manuka honey on *P. aeruginosa* be accounted for solely by its key components?** The transcriptomic effects of manuka honey on *P. aeruginosa* appear to be greater than the sum of its parts, MGO and sugar, although there were many similar changes observed. Treatment with manuka honey induced transcriptional changes resulting in a unique gene expression profile compared to the profiles of *P. aeruginosa* treated with the major components individually (Fig. 1). Hierarchical clustering analysis of RNA-Seq data revealed that the manuka honey gene expression profile was most similar to that of AH and most different from that of MGO alone. The combination of AHMGO was more similar to MH than MGO alone (Fig. 1A), and this is supported by PCA (Fig. 1B).

Analysis of individual gene changes showed that while many genes are significantly differentially expressed (log$_2$FC $>$ 2; $P_{adj} <$ 0.05) across all treatments, MGO treatment resulted in the highest number of differentially expressed genes overall (Fig. 2). This could be due, in part, to the higher concentration of MGO, because using 0.5$\times$ MIC across all treatments meant that the inhibitory effect on cells was the same but the final MGO concentration was different under each condition. There are several genes among the five most up- or downregulated ones common across multiple treatments (MH, MGO, and AHMGO), such as *ahpF*, which was previously identified and is known to play an important role in the response to oxidative species (Fig. 2). The genes *scoA* and *gsp69* were in the ten most significantly differentially expressed genes across multiple treatments (*scoA* in MH, AHMGO, and AH, *gsp69* in MH and AHMGO). The gene *gsp69* encodes a probable oxidoreductase with homology to AKRs in *Escherichia coli* (49). AKRs are capable of detoxifying MGO by reducing methylglyoxal to hydroxyacetone using NADPH as a cofactor (50–53). Only six genes were differentially expressed across all treatments (Fig. 5). These included the PQS quorum-sensing genes *pqsACDE* and the

gene immediately downstream from this operon, *phnA*. All treatments affect the PQS quorum-sensing system (Fig. 4A), and while this is consistent with previous reports (35, 37), it has not been reported for treatment with MGO alone.

In general, manuka honey affected the same processes as AHMGO and MGO but not always to the same extent or in the same direction (Fig. 3). Many of the PseudoCAP categories shown as being affected by honey, including secreted factors, protein secretion, energy metabolism, and DNA replication and adaption, were similarly affected by MGO and AHMGO treatment but not AH (Fig. 3). This suggests that MGO contributed, at least in part, to these gene expression changes. The data also indicate that AH actually has little contribution to the action of manuka honey in terms of the gene expression changes in the biological process and pathways of *P. aeruginosa*, but the sugar component may be necessary to facilitate MGO activity (Fig. 3 and 4).

There were certain categories that were differently affected by manuka honey relative to its major components, for example, genes in the transport of small molecules category were mainly downregulated by manuka honey, whereas AHMGO and MGO seemed to induce both up- and downregulation. The significantly higher number of differentially expressed genes in AHMGO and MGO treatment may be a downstream effect of the higher degree of differential expression of genes in the transcriptional regulators category (Fig. 3), which includes transcriptional regulators such as *lasR*, *rhlR*, *algQ*, and *mvfR* (also known as *pqsR*). All of these genes are global regulators controlling the transcription of large sets of genes across the *P. aeruginosa* genome in response to different stimuli (54).

Like manuka, the expression of genes in the oxidative stress response was also affected by AHMGO and MGO. These data are congruent with the expression data of genes involved in the SOS response, where AHMGO and MGO induced strong upregulation in a wide range of genes involved in SOS, notably *recA* and *lexA* (Fig. 4). MGO is known to cause damage to DNA by modification of guanine bases (25, 55, 56) and has been reported to inhibit the initiation of DNA replication, causing double-stranded breaks in DNA that induce DNA repair (24). MGO treatment has been shown to induce the SOS response in *Bacillus subtilis* (57); therefore, the strong upregulation of genes in the SOS response by both MGO and AHMGO is not surprising.

Curiously, our data show that manuka honey induces expression of SOS response genes comparable to that of MGO- and AHMGO-treated cells despite containing a much lower concentration of MGO. Previous research showed that the expression of SOS genes reduces over time after initial exposure to MGO, and this is thought to be due to the initial transient depletion of glutathione (GSH), which is required for the function of the GSH-dependent glyoxalase systems of *gloA* genes (24). While it is clear that MGO plays a role here, the upregulation of SOS genes by manuka honey cannot be solely attributed to this component.

While similarities across the treatments were seen, we identified 76 genes as being uniquely differentially expressed by manuka honey (Fig. 5). These genes included phage-related genes (Fig. 3), such as the chromosomally encoded tailocin genes *hol* and *lys* (Fig. 4D), which are involved in explosive cell lysis, mediated though the tailocin pathway and dependent on endolysin (*lys*). We also saw significant gene expression changes in the heme oxygenase *nemO* gene, oxidative stress response genes *sodM*, *ankB*, and *katA*, and metabolic genes *fumC1* and *glpD2*. Uniquely downregulated genes include those encoding ABC transporters, *ybeJ*, *gltJK*, and *dppD*, metabolic genes *atoB*, *braE*, *maiA*, *fahA*, and *gnyB*, and the cytoplasmic potassium transporter $K^+$ binding and translocating subunit *kdpA*.

We tested the susceptibility of single-gene knockout mutants ($n = 23$) of *P. aeruginosa* that were either highly or uniquely differentially expressed after manuka treatment (Table S3). One mutant, the Δ*gloA3* mutant (encoding a glyoxalase enzyme for MGO detoxification), showed increased susceptibility (MIC of 5%) relative to that of the wild type (MIC of 10%) (Table S3), suggesting that this gene is important for the survival of *P. aeruginosa* in the presence of honey. The Δ*gloA3* strain is deficient in lactoylglutathione lyase, one of the three redundant enzymes required for the

conversion of MGO and glutathione to lactoylglutatione, suggesting that lactoyl-glutathione lyase plays a role in the antimicrobial action of manuka. However, it is still unclear whether this is due solely to its capacity to detoxify MGO or other downstream effects. The chemical complexity of honey suggests that it targets multiple pathways or proteins; therefore, a single mutation may not lead to a change in MIC. This is consistent with the inability of bacteria to develop resistance to honey (7).

One of the most perturbed pathways for manuka honey-treated cells is related to aerobic respiration, for example, *nemO*, *phuT*, and *phuS* (Fig. 5). PhuST can maintain iron homeostasis by binding heme and either stores it or transfers it to NemO, which then can liberate iron (58). Heme is a cofactor of cytochromes and acts as the electron shuttle for many enzymes in the electron transport chain, playing a critical role in cellular respiration (59). Combined with the expression levels of genes involved in the electron transport chain and central carbon metabolism (Fig. 4E), along with the unique expression of the cytoplasmic membrane depolarizing gene *hol* (60, 61), we hypothe-sized that manuka honey affects the proton motive force (PMF) of *P. aeruginosa*.

**Collapse of the proton motive force: a unique contributor to the antimicrobial activity of manuka honey?** We applied two independent approaches to investigate the impact of manuka on the PMF. To examine directly whether compounds in manuka honey facilitate the passage of protons across biological membranes, we used lipo-somes loaded with the pH-sensitive dye pyranine, allowing the detection of proton movement across the liposome lumen using fluorescence. Liposomes were formed in buffer containing only potassium salts at pH 7.0 and then diluted into buffer containing only sodium at pH 7.0. The addition of a low concentration of the potassium ionophore valinomycin allowed potassium to move down its concentration gradient out of the liposomes, generating an outside positive electrical gradient. The subsequent addition of manuka caused a rapid drop in the internal pH, indicating proton movement into the liposomes (Fig. 6A). The proton movement was dependent on the establishment of an electrical gradient, since no pH change was observed in the absence of valimomycin. No significant change in pH was observed in experiments using AH, AHMGO, or MGO alone, suggesting that a unique component of manuka honey is required. This com-ponent appears to be acting as a novel protonophore that facilitated the passage of protons down an electrical gradient across a biological membrane (Fig. 6A).

To validate whether manuka honey can induce membrane depolarization in live *P. aeruginosa* cells, we used flow cytometry with $DiBAC_4(3)$, a fluorophore subject to selective uptake in depolarized cells (representative plots are shown in Fig. S3). We assessed the number of cells with depolarized membranes after treatment (2 h) with manuka and its key components. Carbonyl cyanide *m*-chlorophenylhydrazone (CCCP; 100 $\mu$M), a PMF uncoupler, was included as a positive control (Fig. 6B). The measure-ment of membrane potential in *P. aeruginosa* is complicated by outer membrane exclusion of fluorophores, and EDTA pretreatment is often used to increase dye uptake in Gram-negative bacteria (62, 63). However, this induced wide-spread membrane depolarization, making negative and positive controls indistinguishable (data not shown). A well-characterized hyperporinated *P. aeruginosa* PAO1 strain expressing a chromosomally located gene for a modified *E. coli* siderophore uptake channel (64) was used to overcome these limitations.

Manuka honey induced significant membrane depolarization in *P. aeruginosa*, unlike MGO and AH (Fig. 6B). While AHMGO induced significant membrane depolarization relative to that of the untreated control, this was at levels significantly lower than those of manuka honey. The exchange of protons across lipid bilayers by manuka honey (Fig. 6A) suggests a dissipation of the PMF and is consistent with our data showing an overall collapse of PMF in *P. aeruginosa* after treatment with manuka (Fig. 6B). However, manuka treatment resulted in an increased number of cells positive for TO-PRO-3 fluorescence, indicating membrane permeabilization. This strongly suggests that the depolarization observed in *P. aeruginosa* cells is a result of damage to the cytoplasmic membrane (Fig. 6C).

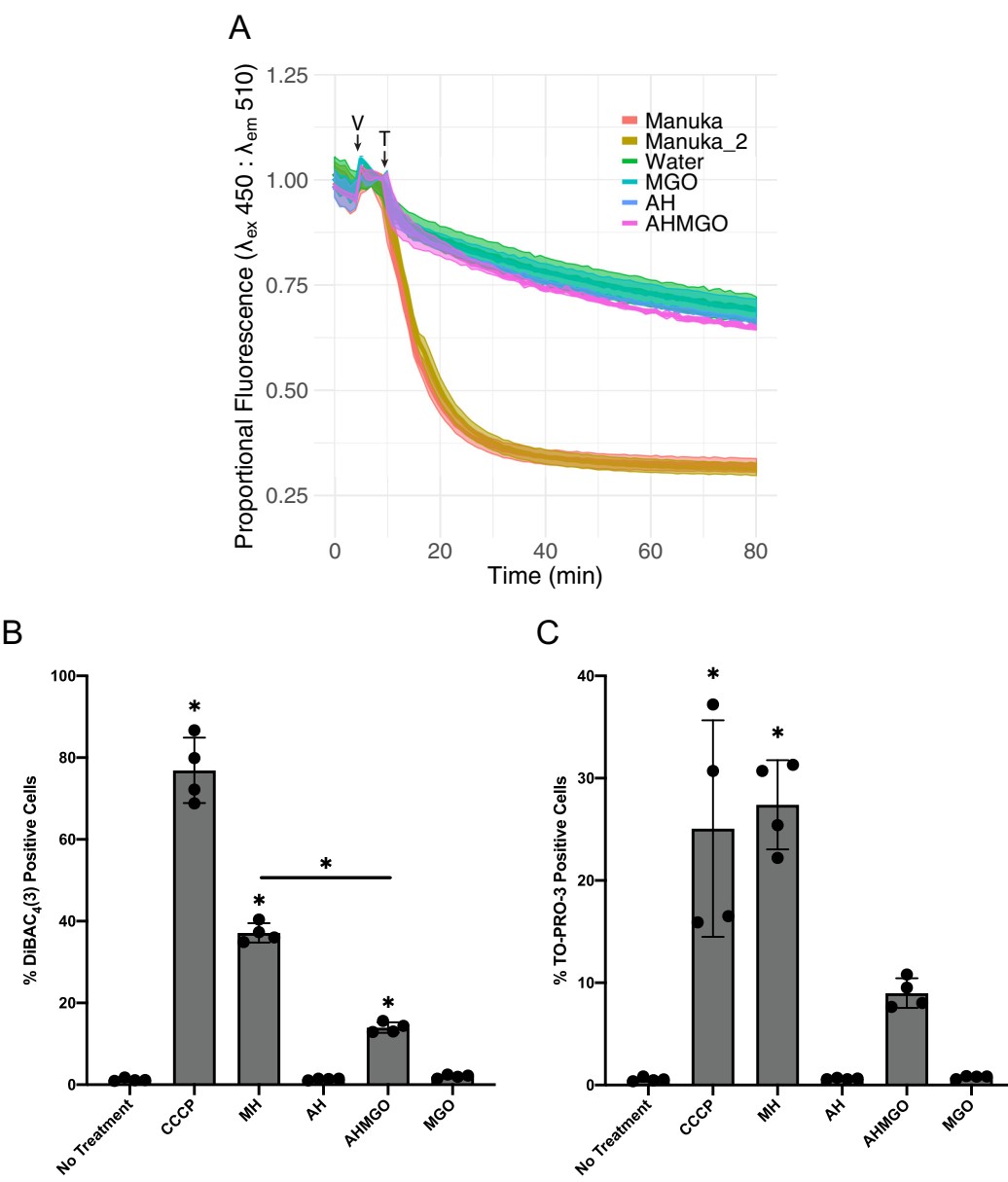

**FIG 6** (A) Effect of manuka honey on membrane potential, measured as approximate internal pH of liposomes loaded with pH-sensitive pyranine dye after treatment with 1%, wt/vol, artificial honey (blue), 1%, wt/vol, MGO (teal), 1%, wt/vol, AHMGO (purple), 1% manuka honey (orange), and a second sample of 1% manuka honey (tan). Fluorescence intensity of pyranine at an excitation wavelength ($\lambda_{ex}$) of 450 nm decreases with decreasing pH. A decrease in proportional fluorescence is indicative of a decrease in internal pH. V indicates the addition of valinomycin to establish an electrical gradient, and T indicates the time at which treatment was applied. (B) Flow cytometry quantification of the percentage of $DiBAC_4(3)$-positive exponential-phase *P. aeruginosa* PAO1-EcPore cells treated for 2 h with 100 $\mu$M CCCP, 10%, wt/vol, manuka honey, 10% AH, 10%, wt/vol, AHMGO, 10% MGO, and a no treatment control. (C) The effect of manuka honey on membrane permeability, measured as flow cytometry quantification of the percentage of TO-PRO-3-positive exponential-phase *P. aeruginosa* PAO1-EcPore cells treated for 2 h with 100 $\mu$M CCCP, 10%, wt/vol, manuka honey, 10% AH, 10%, wt/vol, AHMGO, 10% MGO, and a no treatment control. A one-way analysis of variance (ANOVA) followed by Dunnett's multiple comparison *post hoc* test was used to determine statistically significant differences between each treatment and the no treatment control (*, $P < 0.05$). A one-way ANOVA followed by Bonferroni's multiple comparison *post hoc* test was used to determine statistically significant differences between manuka honey- and AHMGO-treated cells (*, $P < 0.05$).

Because of the observation of membrane permeabilization and depolarization, we hypothesized that manuka honey affects the activity of antibiotics, for example, tetracyclines, to which *P. aeruginosa* is innately resistant. Tetracyclines inhibit the binding of aminoacyl-tRNA to the mRNA translation complex (65), and a major mechanism of

**TABLE 2** Summary of results from checkerboard analysis of combined effects of honey and tetracycline on *P. aeruginosa* PA14 growth[a]

| Antibacterial agent | Antibiotic MIC (μg/ml) | | Honey MIC (%, wt/vol) | | FICI | Synergy (≤0.5) |
|---|---|---|---|---|---|---|
| | Alone | With honey | Alone | With antibiotic | | |
| Tetracycline | 32 | 4 | 10 | 2.5 | 0.37 | Yes |
| Doxycycline | 64 | 8 | 10 | 1.25 | 0.25 | Yes |
| Minocycline | 8 | 2 | 10 | 2.5 | 0.50 | Yes |
| Tigecycline | 4 | 1 | 10 | 5 | 0.75 | |

[a]MIC, minimum inhibitory concentration; FICI, fractional inhibitory concentration index.

tetracycline resistance is through cytoplasmic membrane drug transporters, which require PMF for drug exportation (65, 66). We expect that manuka honey treatment could increase tetracycline uptake due to increased permeabilization and reduce efflux as a result of PMF collapse and, thus, enhance the potency of these antibiotics.

Accordingly, four tetracyclines were chosen for the assessment of synergistic interaction with manuka honey by checkerboard assays, among which tetracycline, doxycycline, and minocycline are known substrates to Tet efflux pumps (TetA/B) (65) and some resistance-nodulation-division (RND) family transporters (MexAB-OprM, MexXY-OprM, and MexEF-OprN) (67, 68), but tigecycline is not recognized by Tet transporters and has a much weaker interaction as a substrate to the RND pumps (69). The functionality of the RND pumps is also membrane potential dependent. Consistent with our hypothesis, apart from tigecycline, manuka honey had strong synergy with the tetracycline antibiotics (Table 2). Furthermore, the synergy is positively correlated with the MICs of the tetracyclines (Table 2), suggesting that manuka honey is able to restore tetracycline antibiotic potency against the bacterial strains that would otherwise be resistant. Tetracyclines have other resistance determinants, such as ribosomal protection proteins and enzymatic inactivation (65). We cannot exclude the possibility that the cause of the tetracycline-manuka synergy is more than membrane depolarization and permeabilization, but this is beyond the scope of the current study.

The PMF is an attractive target for antimicrobial therapy, as it is a fundamental process in energy generation for bacteria. A collapse in the PMF impedes the ability of bacteria to generate energy required to drive processes necessary for resistance to antibiotics, for example, detoxification of tetracycline by PMF-driven multidrug-resistant efflux pumps (66). Our data suggest that manuka honey collapses the PMF in *P. aeruginosa* (Fig. 6) and that this is a biophysically driven mechanism, such as damage to the cytoplasmic membrane, and is independent of proteins involved in the electron transport chain. Previous reports have shown that manuka honey acts synergistically with tetracycline against *S. aureus* (70) but only additively against *P. aeruginosa* (71); however, our data suggest manuka honey also acts synergistically with tetracycline against *P. aeruginosa* (Table 2). *P. aeruginosa* is intrinsically resistant to tetracycline due to drug efflux mediated through PMF-dependent MexAB-OprM and MexXY-OprM RND multidrug efflux pumps (72). The collapse of the PMF would also impair the function of proton-dependent efflux systems, which may explain the synergistic interaction of manuka honey and tetracyclines. These data suggest a role for manuka honey as a therapeutic adjuvant potentially restoring the therapeutic utility of antimicrobials no longer used to treat *P. aeruginosa* infections or open up new treatment options for topical *P. aeruginosa* infections.

**Conclusions.** This study is the first to use a global transcriptomic approach, RNA-Seq, combined with classic microbiology techniques, to investigate the effects and antibacterial mechanism of action of manuka honey and its key antibacterial components against *P. aeruginosa*. We demonstrate that (i) manuka honey induces widespread transcriptional changes and affects many biological processes; (ii) these changes are not wholly explained by its key components, sugar and MGO, either alone or in combination; (iii) MGO, widely accepted as the single most important antibacterial

component of manuka honey, does not account for its total activity against *P. aeruginosa*; and (iv) the collapse of the proton motive force and membrane permeabilization may be a key contributor to the unique antimicrobial activity of manuka honey.

## MATERIALS AND METHODS

**Bacterial strains, media, and antimicrobial agents.** The bacterial strains used in this study are described in Table S1 in the supplemental material. Strains were cultured on cation-adjusted Mueller-Hinton (CAMH) medium grown aerobically at 37°C unless stated otherwise.

The manuka honey used in this study is the same unprocessed honey that was collected and prepared as previously described (20) (MGO, 958 mg/kg; $H_2O_2$, 0.34 $\mu$mol/h), and it was supplied by Comvita Ltd., New Zealand. Artificial honey (AH) was made as sugar solutions of fructose (41.4%, wt/vol), glucose (37.3%, wt/vol), sucrose (2.9%, wt/vol), and water (18.4%, wt/vol) (73) and served as a means to measure the effects of the sugar component of honey. Methylglyoxal (MGO; Sigma-Aldrich)-only treatment was prepared as an aqueous solution at a final concentration equivalent to that of the manuka honey sample (958 mg/kg). This served as a measure of the contribution of MGO relative to that of the manuka honey. An artificial honey doped with methylglyoxal (AHMGO) was prepared per the AH recipe described above, with the modification of adding MGO at a concentration equivalent to that of the manuka honey sample (958 mg/kg) to the water component prior to solubilizing the sugars. AH, AHMGO, and MGO samples were adjusted to pH 4.6 (the native pH of manuka honey) using sodium citrate and then filter sterilized (73). All samples were stored in the dark at 4°C and freshly diluted prior to each experiment. Concentrations are reported as percent weight per volume in this study.

**Determination of MIC, MBC, and synergistic interaction of antimicrobial agents.** Minimum inhibitory concentrations (MICs) of all treatments were determined using the broth microdilution method as previously described (74), with minor changes. CAMH broth was used for all assays, and the final concentration of inoculum was $5 \times 10^5$ CFU/ml. Minimum bactericidal concentrations (MBCs) were determined by inoculating fresh CAMH agar plates from wells of the MIC plates, starting at the MIC and up to the highest concentration tested, with a sterile wooden stick and checking for growth after 24 h of incubation at 37°C. For susceptibility testing under anaerobic conditions, cultures were grown in CAMH broth supplemented with 1% $KNO_3$ (Sigma-Aldrich), and anaerobic conditions were achieved using an Anoxomat II system (Mart Microbiology BV). Antimicrobial interactions with honey were characterized by a standard checkerboard as previously described (75); however, the final inoculum concentration was $5 \times 10^5$ CFU/ml. Synergistic, antagonistic, and no interactions were determined using the fractional inhibitory concentration index (FICI) method, as previously described (76), using the equation $\Sigma FIC = FICA + FICB = (CA/MICA) + (CB/MICB)$, where MICA and MICB are the MICs of drugs A and B alone, respectively, and CA and CB are the concentrations of the drugs in combination, respectively, in all of the wells corresponding to a MIC.

**Total RNA isolation.** *P. aeruginosa* PA14 cultures were prepared in CAMH broth to an initial optical density at 600 nm ($OD_{600}$) of 0.05 and then incubated in a 250-ml cell culture flask (Falcon; Corning) at 37°C with shaking at 200 rpm until reaching mid-exponential phase ($OD_{600}$ of 0.4). Cultures were then split into four flasks containing 2 ml of each treatment at a final concentration of $0.5\times$ MIC: manuka honey (5%, wt/vol), artificial honey (12.5%, wt/vol), artificial honey with MGO (10.75%, wt/vol), and MGO solution (27.5%, wt/vol). A fifth flask containing 2 ml of fresh CAMHB remained untreated. Cultures were grown for an additional 30 min before lysing with QIAzol lysis reagent (Qiagen). Total RNA was isolated using an RNeasy minikit (Qiagen) and DNA removed with DNase I (Turbo DNA-free kit; Invitrogen), as previously described (77) and according to the manufacturers' instructions. Experiments were conducted in triplicate and samples sent to Macrogen (Seoul, South Korea) for rRNA reduction using a Ribo-Zero rRNA removal kit (Illumina), library preparation using the TruSeq stranded mRNA kit (Illumina), and subsequent 100-bp paired-end RNA sequencing on a HiSeq4000 sequencer (Illumina).

**Bioinformatic analysis.** RNA-Seq read quality was assessed using FASTQC (version 0.11.5) and trimmed using Trimmomatic (version 0.36) with default parameters and trimmed of adaptor sequences (TruSeq3 paired-ended). Reads were aligned to the *P. aeruginosa* UCBPP-PA14 genome (http://bacteria.ensembl.org/Pseudomonas_aeruginosa_ucbpp_pa14/Info/Index/, assembly ASM14162v1) and then counted using the RSubread aligner (version 1.30.7) with default parameters (78). After mapping, differential expression analysis was carried out using strand-specific gene-wise quantification using the DESeq2 package (version 1.18.0) (79). Further normalization was conducted using RUVSeq (version 1.13.0) and the RUV correction method, with $k = 1$ to correct for batch effects, using replicate samples to estimate the factors of unwanted variation (80). Absolute counts were transformed into standard z-scores for each gene over all treatments, that is, absolute read for a gene minus mean read count for that gene over all samples and then divided by the standard deviation for all counts over all samples. Genes with an adjusted $P$ value ($P_{adj}$) of $\leq 0.05$ were considered differentially expressed. PseudoCAP analysis was conducted by calculating the percentage of genes in each classification that were differentially expression ($log_2FC \geq \pm 2$, $P_{adj} \leq 0.05$). Classifications were downloaded from the *Pseudomonas* Community Annotation Project (42).

**Assessment of membrane potential after antimicrobial treatment.** Liposomes were formed using *Escherichia coli* polar lipid extract (Avanti Polar Lipids). Lipids were dried under argon or nitrogen from a chloroform suspension to form a lipid film in a glass tube. The lipids were suspended in liposome buffer (25 mM HEPES-NaOH [pH 7.0], 200 mM NaCl, 1 mM dithiothreitol) and subjected to 11 passages of extrusion each through 0.4-$\mu$m and then 0.2-$\mu$m polycarbonate. Five hundred-microliter samples were prepared in the same buffer, including 5 mg of preformed liposomes, 1 mM pyranine (8-hydroxypyrene-

1,3,6-trisulfonic acid trisodium salt), and 1.1% *n*-octylglucoside. The samples were incubated at room temperature for 15 min and then diluted 1:60 with cold liposome buffer to dilute the *n*-octylglucoside to a concentration below its critical micelle concentration. The diluted samples were ultracentrifuged (185,000 × *g*) for 2 h to collect the liposomes that were resuspended in 100 $\mu$l of liposome buffer. For each experiment, liposomes were diluted 1:100 into assay buffer (25 mM HEPES-KOH [pH 7.0], 200 mM KCl), and pyranine fluorescence was continuously monitored to detect pH changes in the lumen of the liposomes [$F_{509}$ (emission at 450 nm)/$F_{509}$ (emission at 400 nm)]. A low concentration (5 nM) of the potassium ionophore valinomycin was added to facilitate the formation of an electrical gradient across the membrane, followed by whole manuka honey or honey components. In control experiments, the polarity of the electrical gradient was reversed by reversing the isosmolar sodium and potassium salts in the liposome and assay buffers.

Membrane potential was assessed using the voltage-sensitive fluorophore DiBAC$_4$(3) and the membrane-impermeable dye TO-PRO-3. Mid-exponential-phase cells (OD$_{600}$ of 0.4) were treated with 10%, wt/vol, of either manuka honey, AH, AHMGO, MGO, or the positive control, 100 $\mu$M CCCP, for 120 min at 37°C with shaking at 200 rpm. Next, 10 $\mu$l of each sample was added to 490 $\mu$l of phosphate-buffered saline containing 0.1 nM TO-PRO-3 (Life Technologies) and 0.5 $\mu$M DiBAC$_4$(3) (Sigma-Aldrich) (final dimethyl sulfoxide concentration did not exceed 1%, vol/vol) and left to incubate at room temperature in the dark for 15 min. Samples then were analyzed on a BD LSRII flow cytometer for forward scatter (FSC), side scatter (SSC), fluorescein isothiocyanate A (FITC-A), and allophycocyanin (APC) fluorescence. A total of 30,000 events were recorded for each sample and gated based on FSC and SSC. Subsequent analysis of flow cytometry data was conducted using FlowJo software (version 10.5.0).

**Data availability.** All data generated and analyzed during this study are included in this article and the supplemental material and were deposited in the Gene Expression Omnibus (GEO) data with the accession number GSE142448.

## SUPPLEMENTAL MATERIAL

Supplemental material is available online only.

**FIG S1**, EPS file, 1.4 MB.
**FIG S2**, PDF file, 0.2 MB.
**FIG S3**, PDF file, 0.5 MB.
**TABLE S1**, DOCX file, 0.02 MB.
**TABLE S2**, DOCX file, 0.02 MB.
**TABLE S3**, DOCX file, 0.02 MB.
**TABLE S4**, DOCX file, 0.02 MB.

## ACKNOWLEDGMENTS

We thank Comvita New Zealand for supplying the manuka honey sample. Flow cytometry was performed at the UTS Microbial Imaging Facility. We also thank Merilyn Manley-Harris for advice on the formulation of artificial honey, Helen Zgurskaya for providing the hyperporinated *P. aeruginosa* strain, Shona Blair for constructive criticism and advice, and Jordana Goth for assistance with knockout screening.

This work was supported by a UTS Doctoral Scholarship awarded to Daniel Bouzo.

D.B., E.H., K.H., L.L., and N.C. contributed to the conception and design of the work. D.B., N.C., E.H., L.L., K.H., G.B., J.L., and A.B. contributed to the acquisition, analysis, and interpretation of data for the work. The paper was written by D.B. and N.C. and critically revised by E.H., L.L., K.H., I.P., A.B., and C.W.

Comvita New Zealand provided materials (honey samples) for the work described in the manuscript. We declare that the research was conducted in the absence of any commercial, financial, or personal relationships that could be construed as a potential conflict of interest.

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
