## [Reviewer comments · mSystems]

Characterising the mechanism of action of an ancient antimicrobial, manuka honey, against *Pseudomonas aeruginosa* using modern transcriptomics

Daniel Bouzo, Nural Cokcetin, Liping Li, Giulia Ballerin, Amy Bottomley, James Lazenby, Cynthia Whitchurch, Ian Paulsen, Karl Hassan, and Elizabeth Harry

Corresponding Author(s): Nural Cokcetin, University of Technology Sydney

Review Timeline:

Submission Date:	February 9, 2020
Editorial Decision:	March 21, 2020
Revision Received:	April 14, 2020
Accepted:	May 15, 2020

Editor: Tricia Van Laar

Reviewer(s): Disclosure of reviewer identity is with reference to reviewer comments included in decision letter(s). The following individuals involved in review of your submission have agreed to reveal their identity: Juraj Majtan (Reviewer #1); Janet Cheruiyot Kosgey (Reviewer #2)

Transaction Report:

DOI: <https://doi.org/10.1128/mSystems.00106-20>

March 21, 2020

Dr. Nural N Cokcetin
University of Technology Sydney
itthree institute
Ultimo, NSW 2007
Australia

Re: mSystems00106-20 (Characterising the mechanism of action of an ancient antimicrobial, honey, using modern transcriptomics)

Dear Dr. Nural N Cokcetin:

We are pleased to accept your manuscript for publication pending minor revisions suggested by the reviewers. The biggest request is an expansion of details in the materials and methods section among a handful of other suggestions. Please address the reviewers' concerns prior to resubmission.

Below you will find the comments of the reviewers.

To submit your modified manuscript, log onto the eJP submission site at <https://msystems.msubmit.net/cgi-bin/main.plex>. If you cannot remember your password, click the "Can't remember your password?" link and follow the instructions on the screen. Go to Author Tasks and click the appropriate manuscript title to begin the resubmission process. The information that you entered when you first submitted the paper will be displayed. Please update the information as necessary. Provide (1) point-by-point responses to the issues raised by the reviewers as file type "Response to Reviewers," not in your cover letter, and (2) a PDF file that indicates the changes from the original submission (by highlighting or underlining the changes) as file type "Marked Up Manuscript - For Review Only."

Please return the manuscript within 60 days; if you cannot complete the modification within this time period, please contact me. If you do not wish to modify the manuscript and prefer to submit it to another journal, please notify me of your decision immediately so that the manuscript may be formally withdrawn from consideration by mSystems.

To avoid unnecessary delay in publication should your modified manuscript be accepted, it is important that all elements you upload meet the technical requirements for production. I strongly recommend that you check your digital images using the Rapid Inspector tool at <http://rapidinspector.cadmus.com/RapidInspector/zmw/>.

Corresponding authors may join or renew ASM membership to obtain discounts on publication fees.

Need to upgrade your membership level? Please contact Customer Service at Service@asmusa.org.

Sincerely,

Tricia Van Laar

Editor, mSystems

Journals Department
Reviewer comments:

Reviewer #1 (Comments for the Author):

Submitted manuscript entitled " Characterising the mechanism of action of an ancient antimicrobial, honey, using modern transcriptomics" has attempted to reveal the mechanism of honey antibacterial action against *Pseudomonas aeruginosa* using transcriptomics approach.

Overall, topic of ms is interesting and worthy to investigate. Authors focused solely on manuka honey and only one bacterial species, namely *Pseudomonas aeruginosa*. Therefore, authors should re-write the title of the ms.

In ms, authors introduced the potential readers about manuka honey and suggested that MGO seems to be only partially responsible for antibacterial activity. Three antimicrobials were tested in study: manuka honey, MGO and artificial honey (sugars only). Antibacterial activity was determined only against planktonic cells and not biofilm-embedded bacteria. It would be interesting to see how manuka honey and its components acts against biofilm of *P. aeruginosa*.

From methodical point of view, authors used modern techniques that were well described in ms. Overall, study was very well conducted and authors found that whole manuka honey is more effective than single active components including MGO.

Reviewer #2 (Comments for the Author):

Specific comments:

1. Why manuka honey was selected for this study? Is there previous literature suggesting high antifungal activity compared to other types?
2. In material and methods; the authors did not report enough details for the methods but cited a

reference, which means one has to refer to the reference for reproductivity.

a. Determination of Minimal Inhibitory Concentration (MIC), Minimum Bactericidal Concentration (MBC) and synergistic interaction of antimicrobial agents' line 123 (135-136), the authors did not mention how FICI values were obtained rather they cited ref 46

b. The quality and quantity of RNA is dependent on the method and reagents used. Total RNA extraction line 147, the authors did not report the kit or reagent used to extract RNA 47..... instead the author would have mentioned the method like... using TRizol as previously described ref or phenol-chloroform or

3. In paragraph 229-302; the authors reported that 'There was no difference in the MIC of manuka honey under aerobic versus anaerobic conditions, suggesting that ROS (and related oxidative stress) is not the only contributor to the antimicrobial mechanism of action', The results justifying this statement is not indicated. This can be demonstrated by adding an additional column in table 2, indicating values of MIC at anaerobic and aerobic conditions, then indicating their p value- no significant difference.

4. The manuscript should be checked for typos; line 750 *P. aeruginosa* is not italicised

Reviewer 1 comments	Point-by-point response to reviewers
Overall, topic of ms is interesting and worthy to investigate. Authors focused solely on manuka honey and only one bacterial species, namely Pseudomonas aeruginosa. Therefore, authors should re-write the title of the ms. In ms, authors introduced the potential readers about manuka honey and suggested that MGO seems to be only partially responsible for antibacterial activity. Three antimicrobials were tested in study: manuka honey, MGO and artificial honey (sugars only). Antibacterial activity was determined only against planktonic cells and not biofilm-embedded bacteria. It would be interesting to see how manuka honey and its components acts against biofilm of P. aeruginosa. From methodical point of view, authors used modern techniques that were well described in ms. Overall, study was very well conducted and authors found that whole manuka honey is more effective than single active components including MGO.	The manuscript title has been changed to include the feedback from Reviewer 1. Original title: Characterising the mechanism of action of an ancient antimicrobial, honey, using modern transcriptomics Amended title: Characterising the mechanism of action of an ancient antimicrobial, manuka honey, against Pseudomonas aeruginosa using modern transcriptomics

Reviewer 2 comments	Point-by-point response to reviewers
1. Why manuka honey was selected for this study? Is there previous literature suggesting high antifungal activity compared to other types?	Our focus in the manuscript is on antibacterial and not antifungal activity, so we have not touched on the antifungal activity here. Manuka honey was chosen specifically for its unusually high levels of antibacterial activity, derived from the floral (plant/nectar) source. Much of the activity of manuka honey come from a factor not found in other honey types and this activity is called non-peroxide activity, predominantly derived from a chemical (MGO) present in the honey. This is explained (with citations) in the manuscript lines 55 – 61, and its antibacterial potency further noted against certain bacteria of interest (with citations) in lines 63 – 66.

2. In material and methods; the authors did not report enough details for the methods but cited a reference, which means one has to refer to the reference for reproductivity. a. Determination of Minimal Inhibitory Concentration (MIC), Minimum Bactericidal Concentration (MBC) and synergistic interaction of antimicrobial agents' line 123 (135-136), the authors did not mention how FICI values were obtained rather they cited ref 46 b. The quality and quantity of RNA is dependent on the method and reagents used. Total RNA extraction line 147, the authors did not report the kit or reagent used to extract RNA 47..... instead the author would have mentioned the method like... using TRizol as previously described ref or phenol-chloroform or	a. Details of how the FICI values were calculated are included in the manuscript – see lines 136 – 139. Synergistic, antagonistic and no interactions were determined using the Fractional Inhibitory Concentration Index (FICI) method as previously described ⁴⁷; using the equation: $\Sigma FIC = FICA + FICB = (CA/MICA) + (CB/MICB)$ where MICA and MICB are the MICs of drugs A and B alone, respectively, and CA and CB are the concentrations of the drugs in combination, respectively, in all of the wells corresponding to an MIC. b. Details of the kits used for RNA isolation are included in the manuscript – see lines 148 – 151. Cultures were grown for an additional 30 minutes before lysing with QIAzol lysis reagent (Qiagen), and total RNA was isolated using the RNeasy mini kit (Qiagen) and DNA removed with DNase I (Turbo DNA-free™ kit; Invitrogen) as previously described ⁴⁸ and according to the manufacturers' instructions.
3. In paragraph 229-302; the authors reported that 'There was no difference in the MIC of manuka honey under aerobic versus anaerobic conditions, suggesting that ROS (and related oxidative stress) is not the only contributor to the antimicrobial mechanism of action', The results justifying this statement is not indicated. This can be demonstrated by adding an additional column in table 2, indicating values of MIC at anaerobic and aerobic conditions, then indicating their p value- no significant difference.	We have included the manuka honey MICs under aerobic versus anaerobic conditions where the results are reported in the manuscript text, and included the p-value to indicate no significant change – see line 306 – 307. There was no difference in the MIC of manuka honey under aerobic (MIC 10 % w/v) versus anaerobic conditions (MIC 11 % w/v) (p > 0.05), suggesting that ROS (and related oxidative stress) is not the only contributor to the antimicrobial mechanism of action.
4. The manuscript should be checked for typos; line 750 P. aeruginosa is not italised	The manuscript has been re-read for typos, and the change to line (now) 753 has been made.

May 15, 2020

Dr. Nural N Cokcetin
University of Technology Sydney
itthree institute
Ultimo, NSW 2007
Australia

Re: mSystems00106-20R1 (Characterising the mechanism of action of an ancient antimicrobial, manuka honey, against *Pseudomonas aeruginosa* using modern transcriptomics)

Dear Dr. Nural N Cokcetin:

Your manuscript has been accepted, and I am forwarding it to the ASM Journals Department for publication. For your reference, ASM Journals' address is given below. Before it can be scheduled for publication, your manuscript will be checked by the mSystems senior production editor, Ellie Ghatineh, to make sure that all elements meet the technical requirements for publication. She will contact you if anything needs to be revised before copyediting and production can begin. Otherwise, you will be notified when your proofs are ready to be viewed.

Sincerely,

Tricia Van Laar
Editor, mSystems

Journals Department
Supplemental Material Table S2: Accept

Supplemental Material Fig S2: Accept

Supplemental Material Table 4: Accept

Supplemental Material Fig S1: Accept

Supplemental Material Table S1: Accept

Supplemental Material Table S3: Accept

Supplemental Material Fig S3: Accept